# Generation and Characterization of a Transgenic Mouse That Specifically Expresses the Cre Recombinase in Spermatids

**DOI:** 10.3390/genes14050983

**Published:** 2023-04-27

**Authors:** Clara Gobé, Côme Ialy-Radio, Rémi Pierre, Julie Cocquet

**Affiliations:** 1Université Paris Cité, INSERM, CNRS, Institut Cochin, F-75014 Paris, France; 2Homologous Recombination, Embryo Transfer and Cryopreservation Facility, Cochin Institute, University of Paris, F-75006 Paris, France

**Keywords:** conditional knockout, spermatogenesis, spermiogenesis, Cre recombinase, *Acrv1*, *SP10*, fertility, early development

## Abstract

Spermiogenesis is the step during which post-meiotic cells, called spermatids, undergo numerous morphological changes and differentiate into spermatozoa. Thousands of genes have been described to be expressed at this stage and could contribute to spermatid differentiation. Genetically-engineered mouse models using Cre/*LoxP* or CrispR/Cas9 are the favored approaches to characterize gene function and better understand the genetic basis of male infertility. In the present study, we produced a new spermatid-specific *Cre* transgenic mouse line, in which the improved *iCre* recombinase is expressed under the control of the *acrosomal vesicle protein 1* gene promoter (*Acrv1-iCre*). We show that Cre protein expression is restricted to the testis and only detected in round spermatids of stage V to VIII seminiferous tubules. The *Acrv1-iCre* line can conditionally knockout a gene during spermiogenesis with a > 95% efficiency. Therefore, it could be useful to unravel the function of genes during the late stage of spermatogenesis, but it can also be used to produce an embryo with a paternally deleted allele without causing early spermatogenesis defects.

## 1. Introduction

A recent report of the World Health Organization (WHO—Infertility Prevalence Estimates, 1990–2021) indicates that one in six people, and one in ten men, suffer from infertility worldwide. The evaluation of male infertility mostly relies on the analysis of sperm parameters, i.e., sperm concentration, morphology, and motility. For sperm concentration below the threshold of 40 million/mL, conception times are known to be increased, and below 1 to 5 million/mL (defined as moderate to severe oligozoospermia), medically assisted reproduction is often needed to conceive a child. This is also the case when sperm motility is affected (i.e., asthenozoospermia) or when the sperm present morphological defects (i.e., teratozoopermia). Azoospermia is defined as the absence of sperm in the semen and can either be obstructive (i.e., when sperm cells are produced but not ejaculated) or non-obstructive. Since the birth of the first “test-tube child” in 1978 following *in vitro* fertilization (IVF), more than eight million children have been born worldwide following the use of assisted reproductive technologies (ESHRE 2018) such as IVF and also Intracytoplasmic sperm injection, in which a single live spermatozoon is directly injected in the oocyte.

Genetic defects are an important cause of male infertility, but in the majority of cases, the causal gene remains unknown, and the infertility is classified as idiopathic. A recent systematic review listing the validated genetic causes of male infertility has identified more than 100 genes [1], but hundreds or more are expected to contribute to the yet still numerous idiopathic cases of male infertility. 

Spermatogenesis is a complex process which lasts approximately 2.5 months in humans and approximately 35 days in mice [2,3]. It is generally divided into three steps: (i) mitotic proliferation of spermatogonia; (ii) meiosis, which consists of chromosome condensation, synapsis, and genetic recombination between homologous chromosomes prior to repartition in haploid daughter cells called spermatids; and (iii) spermiogenesis, the stage during which spermatids differentiate into spermatozoa. Any defect in one of these steps contributes to male infertility. Thousands of genes have been described to be expressed in spermatids [4,5] and are predicted to contribute to the complex genetic program of spermatid differentiation, but, to date, there is no *in vitro* cell model to investigate spermiogenesis. The mouse model remains, therefore, the favored system to define the role of genes at this stage via the use of genetic tools such as Cre/*LoxP* or CrispR/Cas9, strategies that can produce mutation/deletion inducing genetic loss of function (Knock-Out, KO) [6]. A conditional loss of function, achieved using the Cre/*LoxP* system, is sometimes needed to unravel the role of a gene during spermiogenesis, especially when the constitutive KO is embryonically lethal or when the gene is also required at an earlier stage of spermatogenesis. This system consists in producing mice with a transgene expressing the Cre recombinase at a specific stage (under the control of a promoter that is specific to the cell/tissue of interest), in combination with the presence of *LoxP* sites flanking the gene (or part of the gene) of interest. When expressed, Cre excises the region located between the two *LoxP* sites and leads to a gene KO restricted to the cell/tissue where Cre is expressed [7].

In the reproduction field, several *Cre* models have been created to study the role of genes at specific stages of spermatogenesis (see [8] for review), but none of them appear ideal to study spermiogenesis, either because the Cre is expressed in other tissues, such as the brain for *Tspy-Cre* [9], or because Cre expression leads to chromosomal rearrangements and male infertility for *Prm1-Cre* [10]. 

In the present study, we generated and characterized a new spermatid-specific *Cre* line using the promoter of *Acrv1,* the gene encoding the *acrosomal vesicle protein 1*, combined with the improved Cre recombinase, iCre [11]. 

## 2. Materials and Methods

### 2.1. Generation of Transgenic Acrv1-iCre Mice

To establish the *Rosa26^Acrv1-iCre/+^* line (called hereafter *Acrv1-iCre*), the cDNA of *iCre recombinase* gene was inserted under the control of the *Acrv1* promoter sequence described in [12,13]. An hygromycin-resistant cassette, flanked by *flippase* recognition target sites, was introduced as a selection marker downstream of *Acrv1-iCre*. Two homology arms for the *Rosa26* locus were used to insert the construct at this precise locus. The targeting construct was introduced by electroporation into embryonic stem cells from the C57BL/6J mouse strain and selected on plates containing hygromycin. Positive ESCs clones were identified by polymerase chain reaction (PCR) and confirmed by Southern blot analysis with 5′–3′ probes and an internal probe. Homologous recombinant clones were also sequenced and karyotyped. Stem cells carrying the construct were injected into blastocysts from C57BL/6J mice to obtain chimeric mice. After germline transmission, the generated Knock-In *Acrv1-iCre* mice were crossed with the *Flp-deleter* line to eliminate the hygromycin cassette [14]. No deleterious phenotype or health defects were observed in the transgenic males and females that were produced during the approximately 2 years of the project. 

### 2.2. Other Mouse Strains and Genotyping

All animals analyzed in this study were from a C57BL6/J background, and all experiments were performed on (adult) 3- to 5-month-old males. The mice were hosted in a Specific Pathogen-Free (SPF) animal house and were fed ad libitum with a standard diet and maintained in a temperature- and light-controlled room. Animal procedures were approved by the Université de Paris ethical committee (Comité d’Éthique pour l’Expérimentation Animale; registration number CEEA34.JC.114.12, APAFIS 14214-2017072510448522v26).

The genotype of *Acrv1-iCre* mice was assessed with a set of forward (F) and reverse (R) primers (Acrv1-F: 5′–ACCTGTTCAATTCCCCTGCA–3′; Acrv1-R1: 5′–GTGAAGTGTGGTGACCTGGT–3′; Acrv1-R2: 5′–AGCACGTTTCCGACTTGAGT–3′) which allows the detection of the *Acrv1-iCre* allele (F + R1 produce a 301 bp fragment) and the *Rosa26* WT allele (F + R2 produce a 429 bp fragment). The PCR was conducted as follows: denaturation at 98 °C for 5 min, then 30 times the cycle at 98 °C for 20 s, 60 °C for 30 s, 72 °C for 45 s, and a final extension at 72 °C for 5 min. The *Dot1l* mouse line was genotyped as described in [15] with a set of primers (Dot1l-F: 5′–GCAAGCCTACAGCCTTCATC–3′; Dot1l-R1: 5′–CACCGGATAGTCTCAATAATCTCA–3′; Dot1l-R2: 5′–GAACCACAGGATGCTTCAG–3′). Briefly, primers F + R1 amplify the WT allele at 536 bp and the floxed allele at 642 bp, while primers F + R2 amplify the WT allele at 960 bp, the flox allele at 1017 bp, and the deleted allele (∆) at 335 bp. For an internal control, we used primers amplifying the *Ymtx* gene (Ymtx-F: 5′–TCACACAGATAAGAGGGTATTG–3′; Ymtx-R: 5′–GTTTTCCTATCAGGCCATCCA–3′).

To verify iCre recombinase expression and activity, the *Acrv1-iCre* line was crossed with the reporter line *Rosa26^mTmG/+^* line (hereafter called *mTmG*), described in [16], or with the *Dot1l* floxed line (*Dot1l^Fl/Fl^*), first described in [17]. In these mice, *Dot1l* exon 2 is flanked by *LoxP* sites. Cre-mediated recombination of *LoxP* sites leads to the deletion of exon 2 (See also [15]). From the first type of mating, *Rosa26^mTmG/Acrv1-iCre^* and *Rosa26^mTmG/+^* males were produced and analyzed. From the second type of mating, *Dot1l^Fl/Fl^*; *Rosa26^Acrv1-iCre/+^* (hereafter called *Dot1l* KO), *Dot1l^Fl/Fl^*; *Rosa26^+/+^* control siblings (hereafter called CTL), and *Dot1l*^Fl/+^; *Rosa26^+/+^* heterozygous siblings (hereafter called HTZ) were produced. In parallel, animals with an already deleted *Dot1l* allele (∆) were produced: *Dot1l^Fl/∆^*; *Rosa26^Acrv1-iCre/+^* (KO∆) and *Dot1l^Fl/∆^*; *Rosa26^+/+^* heterozygous siblings (hereafter called HTZ∆).

### 2.3. Western Blot Analysis of Cre Expression

Proteins from the brain, spleen, liver, lung, kidney, heart, ovaries, uterus, and testes were extracted using a 25 mM RIPA buffer (25 mM NaCl, 10 mM Tris-HCl pH7.5, 5 mM EDTA, 0.1% NP-40, 1X PIC, 1 mM PMSF, 5 mM NaBut). Proteins were quantified by Bradford Chromatography Assay (Micro BCA Protein Assay Kit, Thermoscientific, #23235, Waltham, MA, USA) and ~60µg of proteins were loaded. Electrophoresis was performed in polyacrylamide gels at 120 V in a denaturation buffer containing 25 mM Tris, 190 mM glycine, and 0.1% SDS, and proteins were transferred on nitrocellulose membranes (GE Healthcare, Chicago, IL, USA). Membranes were then rinsed and incubated 3 min in Ponceau stain to visualize the proteins after transfer. Then, membranes were incubated in 1X PBS, 0.01% Tween, and 5% milk. Anti-Cre primary antibody (CST #15036, 1/500) and anti-TUBULIN primary antibody (Upstate #05-661, 1/5000) were incubated over night at 4 °C, and the secondary antibody HRP-conjugated (ThermoFisher anti-rabbit or anti-mouse HRP, #31460 or #31430, 1/5000) was incubated 2 h at room temperature. The revelation was performed with SuperSignal West Pico Plus^®^ ECL from ThermoFisher (#34580) and Immobilon ECL Ultra Western HRP substrate from Millipore (WBULS0100). Signals were detected using ImageQuant^™^ LAS 4000 imager.

### 2.4. Immunofluorescence on Testis Section

Mouse testes were fixed in 4% buffered paraformaldehyde (PFA) for a minimum of 4 h, before they were cut in halves and incubated overnight in 4% PFA. The testes were then washed in 70% ethanol at room temperature for 30 min, dehydrated, and embedded in paraffin. Immunohistochemistry or immunofluorescence experiments were performed on 4 µm sections, as described in [15,18]. Primary antibodies were incubated overnight at 4 °C (anti-Cre 1/100, CST #15036; anti-GFP 1/100, ThermoFisher #CAB4211). For immunofluorescence experiments, LECTIN (LECTIN-PNA-488 #L-21409 or LECTIN-PNA-594 #L-32459, ThermoFisher) was diluted at 1/400 in 1X PBS/0.1% BSA and incubated for 1 to 2 h at room temperature along with secondary antibodies (goat anti-rabbit 488 or donkey anti-rabbit 594). LECTIN was used to stain the developing acrosome and determine the stage of testis tubules as described in [2]. DAPI (in VECTASHIELD Mounting Medium, #H-1200, Vectorlab) was used to stain nuclei. All immunofluorescence pictures were taken with an Olympus BX63 microscope and analysed using ImageJ 1.52a (http://imagej.nih.gov/ij/ (accessed on 23 April 2018)).

### 2.5. Sperm Collection and Genomic DNA Extraction

Spermatozoa were extracted from cauda epididymis in pre-warmed (37 °C) M2 medium (Sigma, #SLCN7467, St. Louis, MO, USA) by gentle pressure. Then, epididymides were perforated with a thin needle and incubated in M2 for 5 min at 37 °C to allow remaining sperm to swim up. For molecular analyses, spermatozoa were washed with PBS-BSA (1X PBS, 0.5% BSA, 2 mM EDTA) and centrifuged at 2000× *g* for 10 min. Sperm cell pellets were snap-frozen in liquid nitrogen and stored 6 months at −80 °C prior to use.

Sperm genomic DNA (gDNA) was extracted following the procedure already described in [19]. Briefly, approximately 5 million spermatozoa were incubated overnight at 55 °C in a lysis buffer (Tris–HCl 25 mM, pH 8; SDS 1%, EDTA 5 mM, DTT 0.1 mM, proteinase K 0.4 mg/mL, β-mercaptoethanol 2%). DNA lysates were phenol/chloroform-extracted, EtOH-precipitated (0.1 volume NaAc 3M pH 5.2 with 2 volumes EtOH 95%), and re-suspended in TE buffer (Tris 10 mM, pH 7.5; EDTA 1 mM). DNA samples were digested with RNase A (20 µg/mL) at 37 °C for 20 min, and DNA concentrations were assessed using Thermo Scientific Nanodrop.

### 2.6. Assessment of Acrv1-iCre Recombination Efficiency

*Acrv1–iCre* recombination efficiency was assessed on spermatozoa from adult (3- to 6- month-old) *Dot1l^Fl/Fl^* or *Dot1l^Fl/∆^* males carrying (*Rosa26^Acrv1-iCre/+^*) or not *Acrv1-iCre* transgene (*Rosa26^+/+^*). Genomic DNA was extracted from epididymal spermatozoa and used to assess the efficiency of Cre recombination by semi-quantitative PCR (semi-qPCR) and real-time quantitative PCR (qPCR). For semi-qPCR, 300 ng of gDNA were amplified by PCR for 25 cycles then run on a 1.5% agarose gel. Real-time PCR was performed using the Roche LightCycler 480 and SYBR SensiFAST 2X (BIO-98050, Bioline). For qPCR, 5 ng of gDNA were mixed with SYBR 2X and Dot1l F/R1 or F/R2 primers, then amplified with a pre-incubation 95 °C for 5 min and amplification at 95 °C for 10 s, 55 °C for 15 s, and 72 °C for 15 s for 49 cycles on a LightCycler480. Primers amplifying a fragment of the *Sox17* gene were used as an internal control (Sox17-F: 5′–TTGCTTAGCTCTGCGTTGTG–3′; Sox17-R: 5′–GCCGATGAACGCCTTTATGG–3′).

## 3. Results

### 3.1. Acrv1-iCre Expression Is Testis-Specific and Restricted to Round Spermatids

In order to generate a new spermatid-specific Cre model, we produced a mouse line that expresses the *iCre* [11] under the control of the promoter of the spermatid-specific gene *Acrv1* (also known as *SP10*). *Acrv1* has been shown to be transcribed in round spermatids from seminiferous tubules stages I to VI [13]. Analysis of RNA-seq data on germ cells purified at different stages of spermatogenesis [20] confirmed that it is only expressed in round and elongating spermatids (Figure 1a). We used the region previously described in [12,13] which corresponds to approximately 430 bp upstream of the *Acrv1* transcription start site. This promoter has been shown to drive strong and spermatid-specific expression of other genes, such as the gene coding for the *green fluorescent protein* (GFP) or the gene *Sly* [13,21]. Here, we cloned this promoter sequence upstream of the coding region of *iCre* and added two arms of homology with the *Rosa26* locus (Figure 1b) to insert the construct by homologous recombination at this site. The construct was injected in C57BL/6 embryonic stem cells, and chimeric mice were crossed until germ line transmission of the modified allele. *Acrv1-iCre* transgenic animals were crossed with a *flippase* line to remove the selection cassette (Figure 1c). The *Acrv1-iCre* line was then maintained on a C57BL6 background and genotyped as shown in Figure 1d. No fertility defect was observed in transgenic males or females.

To control that the Cre expression is not leaky, we analyzed by western blot transgenic male and female organs important for survival and/or metabolism, such as the brain, liver, lung, kidney, spleen, and heart, and organs from the male and female reproductive tract, i.e., testes, ovaries, and uterus. As shown in Figure 2, Cre protein is only detected in the testis.

We next checked in which testicular cells Cre protein is expressed by performing immunofluorescence assays using anti-Cre antibody on testicular sections. Cre protein was only detected in step 5 to 8 round spermatids (Figure 3a), with the highest signal intensity in spermatids from stage VI-VII tubules (Figure 3b and Appendix A). No signal was observed in other testicular cells, indicating that Cre is specifically expressed at the spermatid stage in the *Acrv1-iCre* line. Non-specific staining was observed in the intertubular islets from both *Acrv1-iCre* transgenic and non-transgenic testes (Figure 3a). 

### 3.2. Acrv1-iCre Line Is Efficient to Knock out Genes during Spermiogenesis

Despite the fact that Cre expression is specific, this does not ensure its efficiency. We therefore assessed *Acrv1-iCre* efficiency by two different approaches. First, we used the Cre reporter model *mTmG* (i.e., *Rosa26^mTmG/+^*) previously described in [16]. In this model, the GFP is only expressed when Cre has efficiently cut and removed the *Tomato* cassette. We generated *Rosa26^mTmG/Acrv1-iCre^* males that carry both *Acrv1-iCre* and *mTmG* transgenes and detected the GFP by immunofluorescence assays on testicular sections. The results show that the GFP is strongly and specifically expressed in step 10 to 16 elongated/condensed spermatids. No GFP was observed in the testis from negative controls, i.e., *Rosa26^mTmG/+^* males which do not carry the *Acrv1-iCre* transgene (Figure 3a,b). The delay between Cre expression and GFP expression was quite surprising but may be due to the time needed for the *Gfp* gene to be expressed. Moreover, in the *mTmG* line, the GFP is addressed to the membrane, and this might delay the observation of the GFP signal. Secondly, we tested *Acrv1-iCre* efficiency by crossing the *Acrv1-iCre* line with a floxed mouse line to quantify the proportion of deleted (∆) vs. intact (floxed) alleles. For this, we used the *Dot1l* floxed model [15,17] and produced males of five different genotypes: KO animals (*Dot1l^Fl/Fl^*; *Rosa26^Acrv1-iCre/+^*), CTL siblings (*Dot1l^Fl/Fl^*; *Rosa26^+/+^*), HTZ siblings (*Dot1l^Fl/+^*; *Rosa26^+/+^*), KO∆ animals (*Dot1l^Fl/∆^*; *Rosa26^Acrv1-iCre/+^*), and HTZ∆ siblings (*Dot1l^Fl/∆^*; *Rosa26^+/+^*). We extracted the genomic DNA of their epididymal spermatozoa and performed semi-quantitative and quantitative PCR (Figure 4). By semi-quantitative PCR, we tested two KO animals and six KO∆ animals. One KO exhibited a weak signal for the floxed allele, whereas none of the KO∆ animals showed any amplification of the floxed allele (Figure 4b). By qPCR assay on the same samples, we precisely quantified Cre recombinase efficiency. As shown in Figure 4c, in KO animals where the two alleles of *Dot1l* are floxed, we observed that only 3% and 7% of the floxed allele signal remained, the 7% corresponding to the same animal in which a fainted band was detected by semi-qPCR. In KO∆ animals in which one allele was already deleted and only one floxed allele remained, we observed that this percentage was down to 1–3%. Thus, in the *Acrv1-iCre* line, Cre recombinase efficiency is approximately 95% and, in conditions where one allele is already deleted, it is >97% efficient.

## 4. Discussion

In the reproduction field, several *Cre-recombinase*-expressing mouse lines have been created to spatially and temporally control genetic manipulations in the male germ line. Among the most popular models available to date are *Stra8-iCre*, *Vasa-Cre* and *Ngn3-Cre* lines [23,24,25]. *Stra8-iCre* is specifically expressed in postnatal male germ cells, starting in spermatogonia and reaching its maximum efficiency in primary spermatocytes [23]. Despite the use of a new improved Cre recombinase [11], this model has been shown in several instances to be approximately 80% efficient and to generate a knock-down rather than a KO [15,26,27]. *Vasa-Cre* is active in late primordial germ cells during embryonic development and has proven to be more efficient. However, this *Cre* model can only be transmitted by young males (younger than 7 weeks old) to avoid Cre accumulation in spermatozoa and its leakage in the zygote, which makes this model difficult to use [28]. The other model which allows efficient genetic recombination in early male germ cells is the *Ngn3-Cre*. It is largely used in the male reproduction field but has been reported to be also expressed in the brain [29], which could be problematic when targeting male reproduction genes that are also expressed in this organ. In all above-described models, *Cre* expression, combined with the *floxed* alleles of the gene of interest, leads to a pre-meiotic conditional KO. They are therefore useful to investigate the role of genes in spermatogonia during meiosis but cannot address a post-meiotic role, if the conditional KO leads to early germ cell loss or a meiotic block. To investigate the role of genes in post-meiotic male germ cells, two other *Cre* lines expressed in spermatids have been created, but one is not specific to the male germline (i.e., *Tspy-Cre* is also expressed in the brain [9]), and the other induces excision outside of *LoxP* sites and chromosomal rearrangements (*Prm1-Cre*; see [10,30]).

In the present study, we described the generation and characterization of a novel spermatid-specific *Cre* mouse model, *Acrv1-iCre* (see Figure 5). Using qPCR on spermatozoa, we demonstrated that *Acrv1-iCre* is very efficient (about 95 to 97%) to conditionally knock out a gene during spermiogenesis. We showed that the Cre protein is not expressed in any organs other than the testis and is highly and specifically expressed in round spermatids from step 5. Using the *mTmG* reporter line, we detected GFP expression (induced by Cre activity) later than expected based on the Cre pattern of expression. This delay may be due to the time needed for *Gfp* to be transcribed and translated and/or for GFP to be addressed to the membrane; we predict that a gene knockout could be induced earlier, around the time of Cre high expression in step 6 spermatids. It will therefore be too late for genes with a biological role in early round spermatids or for genes encoding stable proteins (i.e., with a long half-life). However, the *Acrv1-iCre* line will be useful to characterize the role of genes required for the last stage of spermatogenesis, since it can bypass early spermatogenesis defects or bias. Finally, the *Acrv1-iCre* line could be a very useful tool for those studying gamete interaction and early embryonic development since it will be possible to generate KO spermatozoa with a high efficiency and transmit a paternally deleted allele to the embryo. All in all, the use of this novel model could help in identifying the genetic causes of defects in late sperm differentiation, in fertilization, or in early embryo development. 

## Figures and Tables

**Figure 1 genes-14-00983-f001:**
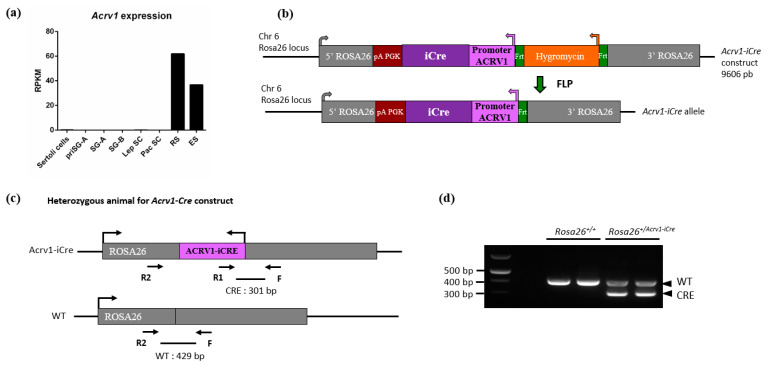
Generation and genotyping of *Acrv1-iCre* animals. (**a**) Representation of *Acrv1* expression pattern in testiscular cells using RNAseq data from [20]. *Acrv1* is only expressed in germ cells, from round spermatids to elongated spermatids. PriSG A: Primitive type A spermatogonia; SG A (or B): Spermatogonia A (or B); Lep SC (or Pac SC): Spermatocytes at Leptotene (or Pachytene stage); RS: Round spermatids; ES: Elongated spermatids. (**b**) Linear representation of the *Acrv1-iCre* construct which was injected in embryonic stem cells. *Acrv1* promoter has been used to drive *iCre* expression; the construct size is 9606 bp. After germline transmission, the first generation of mutant KI animals was crossed with a *flippase* expressing mouse line in order to remove the hygromycin cassette. (**c**) Simplified schematic view of *Acrv1-iCre* heterozygous animals with primer localization. Primers F + R1 + R2 detect both WT and *Acrv1-iCre* alleles. (**d**) Visualization on an agarose gel of the PCR products following genotyping of *Rosa26^+/+^* and *Rosa26^Acrv1-iCre/+^* animals. In gDNA from wild-type (WT) animals, only one amplicon corresponding to the *Rosa26* locus without the *iCre* insertion is visible at 429 bp (amplified with F + R2 primers). For heterozygous (HTZ) animals in which the *Acrv1-iCre* allele is present in one copy, two bands are detected: the WT fragment at 429 bp and a smaller fragment at 301 bp which corresponds to *iCre* detection (amplified with F + R1 primers). NB. F + R2 primers are too far apart on *Acrv1-iCre* allele to produce an amplicon.

**Figure 2 genes-14-00983-f002:**
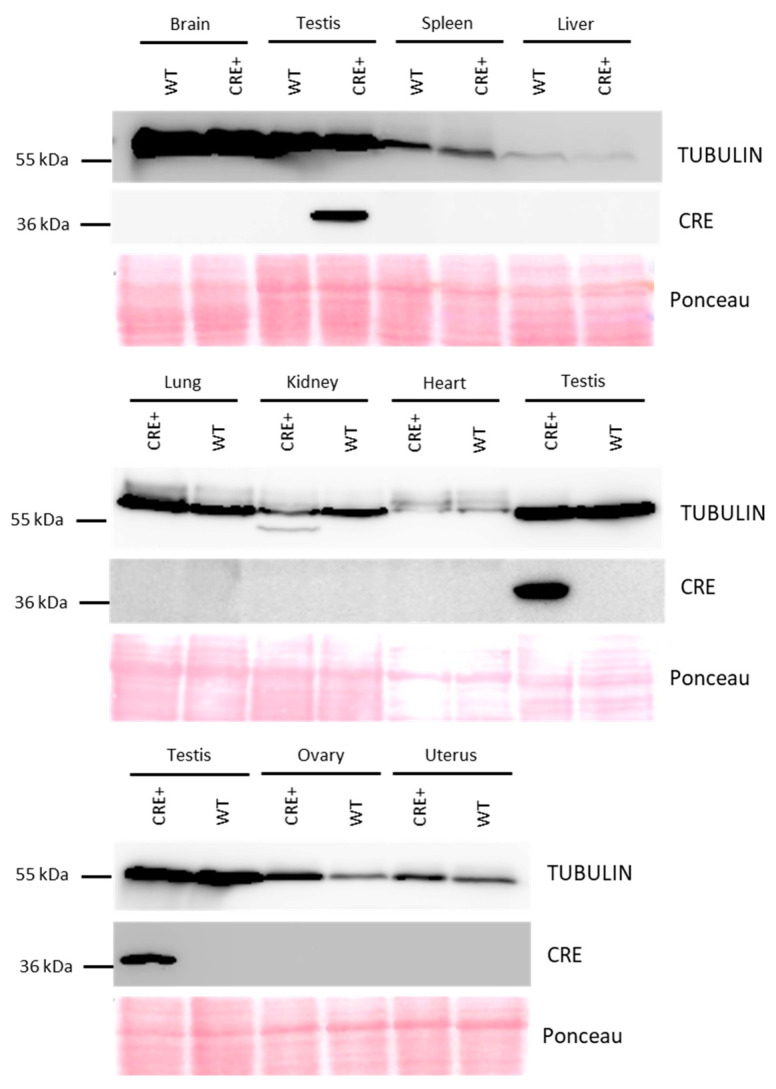
Detection of Cre protein in multiple organs of *Acrv1-iCre* transgenic animals by western blot. Cre protein (detected at 37 kDa) is only found in the testicular extract of *Rosa26^Acrv1-iCre/+^* transgenic animals (CRE+) and not in their negative control siblings *Rosa26^+/+^* (WT). All other organs tested, i.e., brain, spleen, liver, lung, kidney, heart, ovaries, and uterus are negative for iCre protein detection. TUBULIN, at 50 kDa has been used as a loading control. Ponceau stain confirms that the same quantity of proteins was loaded in all the lanes.

**Figure 3 genes-14-00983-f003:**
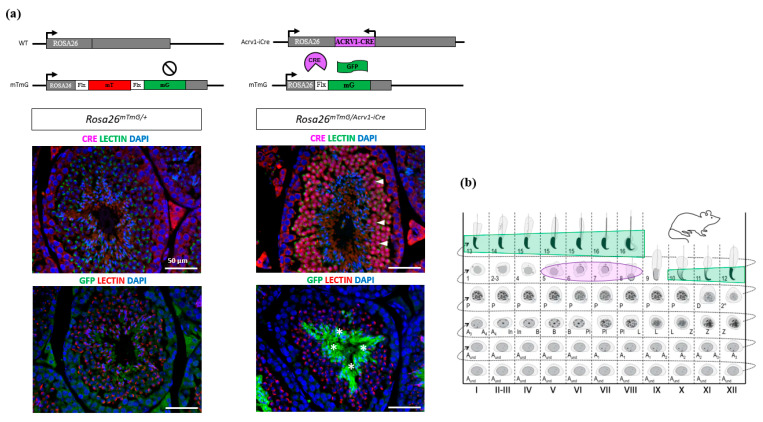
Immunodetection of Cre and GFP proteins on testicular section of adult *Acrv1-iCre* and *mTmG* animals. (**a**) Left panel: in *Rosa26^mTmG/+^* testes which do not express *Acrv1-iCre*, no signal can be detected with anti-Cre antibody, and only LECTIN signal used to stage the seminiferous tubules is visible (photos of stage VI tubules). Likewise, no fluorescent staining for GFP is detectable. Right panel: in *Rosa26^mTmG/Acrv1-iCre^* animals, Cre protein is only present in round spermatids (arrow head) and not in other cells. When Cre is expressed, it removes the *Tomato* cassette and enables GFP expression. In our model, GFP is detected in late elongated/condensed spermatids (*). DAPI is used to stain the nuclei. Scale bar indicates 50 µm. (**b**) Schematic representation of the detection of Cre (in red) and GFP (in green) in the different stages of seminiferous tubules adapted from [22]. Cre protein signal was the strongest in step 6–7 spermatids, while GFP protein signal was the strongest in step 15–16 spermatids. See Appendix A.

**Figure 4 genes-14-00983-f004:**
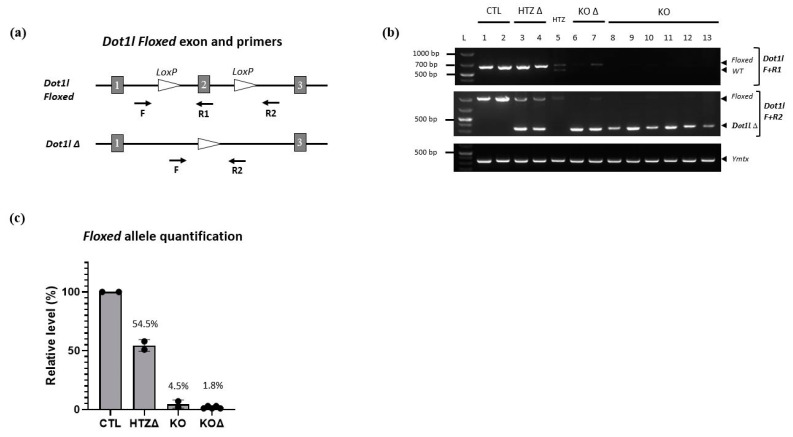
Assessment of *Acrv1-iCre* efficiency by semi-quantitative and quantitative PCR. (**a**) Simplified schematic representation of *Dot1l* floxed exon 2 with localization of the primers used in the present study. Primers F + R1 detect the *floxed* alleles; while F + R2 primers detect the deleted alleles (*∆*). (**b**) Detection of the *floxed*, *wild-type*, or *∆* alleles in *Dotl1* CTL, HTZ∆, HTZ, KO or KO∆ animals by PCR shows that only the floxed fragments are amplified in CTL animals. As expected, both *Dot1l floxed* and *∆* alleles are detected in HTZ∆ animals, and a WT band can be seen for HTZ. In KO and KO∆, the *∆* allele is present; however, a faint *floxed* band can be seen for one KO animal (#7), which is not the case for the other KO∆ animals. (**c**) Assessment of *floxed* allele quantity by real time qPCR. The panel shows mean values (±standard deviation) of *floxed* allele relative levels (normalized to internal *Sox17* values and then to CTL values, set at 100%). As expected, *floxed* allele level is reduced to ~50% in HTZ∆ animals. In KO and KO∆ animals, few or no *floxed* alleles remain.

**Figure 5 genes-14-00983-f005:**
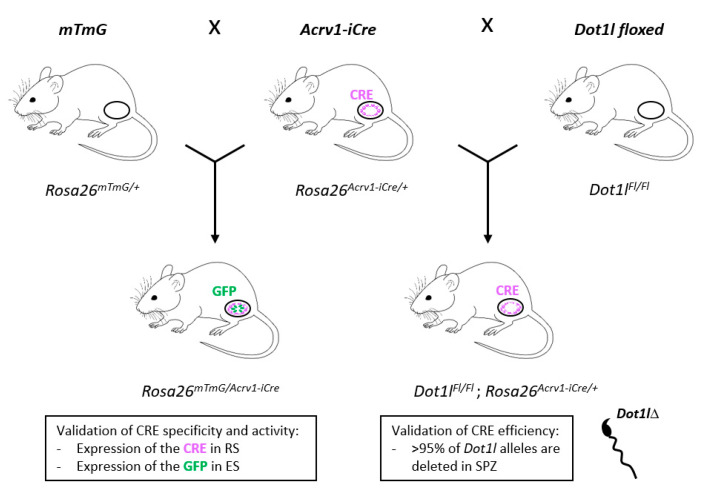
Summary of the results obtained with the *Acrv1-iCre* mouse line. The three mouse models used in the present study (i.e., *mTmG*, *Acrv1-iCre,* and *Dot1l floxed* lines) are shown on the top of the figure; crossings are indicated with arrows; the resulting mice and genotypes are shown underneath. Oval shapes represent the testes. Cre or GFP expression is indicated in purple or green, respectively. RS: Round spermatids; ES: Elongated spermatids; SPZ: Spermatozoa.

## Data Availability

Not applicable.

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
