# Peer review of "Generation and Characterization of a Transgenic Mouse That Specifically Expresses the Cre Recombinase in Spermatids"

_genes, 2023, doi:10.3390/genes14050983_

Round 1
Reviewer 1 Report
Generation and characterization of a transgenic mouse which specifically expresses the Cre recombinase in spermatids- 2280760
In this study, authors reported the generation and characterization of a new spermatid-specific Cre transgenic mouse line, in which newly generated cre recombinase is expressed under the control of the acrosomal vesicle protein 1 gene (Acrv1-icre). This is an interesting study, which is indeed needed in the field of male reproductive medicine. However, for proper reproducibility, some processes require clarification. Herewith are my suggestions to improve the manuscript.
Specific comments
How long were the animals monitored, to ensure that the new line is phenotypically normal/ healthy and that there is no compromise in other organs function.
A brief detailed phenotypic analysis performed after the generation of the new line should be described in the M&M section.
What methods or assays were done to ascertain that mTmG selectively identifies Acrv1?
How was Dot1l line produced in the first place? I know a reference has been cited, but it is easier to comprehend if a brief protocol is stated. What are the characteristics of this line? Is this the line produced after Rosa26 was deleted/floxed?
Beware of line name presentation
A summary figure that will visually showcase the entire process of generation and characterization is needed.
The protocol should be better optimized to allow for exclusive spermatid specific Cre protein expression as this will dispute/ cancel the possibility of Cre leakage into other cells of the testes apart from the spermatids which are the cells of interest. What I am saying is, leakage was tested in other vital organs such as heart, lungs, kidneys, etc but different spermatogenic cells (apart from round spermatids) were not tested.
Line 141- ‘Sperm cells pellets were snap-frozen in liquid nitrogen and stored at -80°C prior to use’- How long were the spermatozoa frozen for? Because gDNA concentration reduces with time and if the start material is bad then the subsequent experiments are unreliable. To avoid doubts, the specific time of storage should be mentioned.
Section 2.6-Assessment of Acrv1-iCre recombination efficiency. The procedure used to isolate different spermatogenic cells should be briefly discussed.
Section 4- Discussion- What the authors described here is not a discussion, it's rather a conclusion of the study. I know few studies are available on the conditional knockout/knockin cre-lox system to study spermatogenesis, yet a proper discussion of the results should be performed. Relate your findings with other studies and highlight the differences and why your line has almost 95% efficiency in knockin.
Line 227- ‘in round spermatids from stage V’ - Your results in the current state did not specifically show that CRE protein is exclusively expressed in the early spermatids. If you are referring to your IF study, proper micrographs showcasing these specific findings should be added to the manuscript.
The entire manuscript should be proofread to avoid the misuse of prepositions.
General comments
Line 10- striking is not scientific
Line 19 - Therefore, it should be.. versus Therefore, it could be
Line 20-21 - but also to produce embryo with a paternally deleted allele without causing early spermatogenesis defects… versus but can as well be used to produce embryo with a paternally deleted allele w/o causing early spermatogenesis defects.
Line 25- ‘Genetic defects are expected to be an important cause of male infertility’ - Not a proper sentence. Genetic defects are important cause of male infertility... they are not expected. Rephrase
stages of spermatogenesis in mice 1. proliferation of spermatogonia, 2. meiotic division of spermatocytes, 3. morphological change of spermatids.
Line 33- (ii) meiosis, which consists in chromosome- which consists of chromosome
Line 36- Any defect in one of these steps leads to male infertility - Any defect in one of these steps contribute to male infertility
Line 44- Insert full stop after spermiogenesis and start a new sentence. For instance, ....
OR revise the entire sentence along these lines:
A conditional loss of function achieved using CRE/LoxP system is sometimes needed to unravel the role of a gene during spermiogenesis, especially for instance when the constitutive KO is embryonically lethal and/or when the gene is also required at an earlier stage of spermatogenesis.
Line 47- (under the control of a promoter which is specific of to the cell/tissue of interest).
Line 51- In the reproduction field, several Cre models have - if it is several, then more than one reference should be cited.
Line 67- Why is figure 1b appearing before figure 1a. Figures should be numbered chronologically in the way of appearance. figure 1b- line 67; figure 1a- line 169; another figure 1b- line 174. Revise the numbering of figures appropriately.
Line 69 - Mention what appropriate means here. ESCs clones?
Line 87- Define terms before use since they are not standardized abbreviations. What is F, R1, R2. be clear
Line 94 - allele at 642 bp, when primers - versus while?
Line 108 - Proteins were extracted- From which organ/tissue/cell were proteins extracted?
Figure 3- Individual images for CRE, GFP and Dapi should likewise be provided, rather than showing only the merged micrograph.
Use arrow or symbols to show the findings reported in line 221-223
Figure 3b- What does the I - XII connote? In addition to the semi-table representation, show these findings with bigger magnification or better resolution
Figure 1b:
![]()

Shouldn’t this Frt (green box) be removed? Since the region has been floxed

Reviewer 2 Report
- I suggest minor spell check and punctuation.
- The introduction: the authors need to add more information regarding reproductive health problems globally.
- MM and results are very well presented.
- the discussion is very small compared to the presented results
